# Shadow enhancers mediate trade-offs between transcriptional noise and fidelity

**Alvaro Fletcher[1], Zeba Wunderlich[2,3]\*, German Enciso[4]\***

**1** Mathematical, Computational, and Systems Biology, University of California, Irvine, Irvine, CA, United States of America, **2** Department of Biology, Boston University, Boston, MA, United States of America, **3** Biological Design Center, Boston University, Boston, MA, United States of America, **4** Department of Mathematics, University of California, Irvine, Irvine, CA, United States of America

\* zeba@bu.edu (ZW); enciso@uci.edu (GE)

## Abstract

Enhancers are stretches of regulatory DNA that bind transcription factors (TFs) and regulate the expression of a target gene. Shadow enhancers are two or more enhancers that regulate the same target gene in space and time and are associated with most animal developmental genes. These multi-enhancer systems can drive more consistent transcription than single enhancer systems. Nevertheless, it remains unclear why shadow enhancer TF binding sites are distributed across multiple enhancers rather than within a single large enhancer. Here, we use a computational approach to study systems with varying numbers of TF binding sites and enhancers. We employ chemical reaction networks with stochastic dynamics to determine the trends in transcriptional noise and fidelity, two key performance objectives of enhancers. This reveals that while additive shadow enhancers do not differ in noise and fidelity from their single enhancer counterparts, sub- and superadditive shadow enhancers have noise and fidelity trade-offs not available to single enhancers. We also use our computational approach to compare the duplication and splitting of a single enhancer as mechanisms for the generation of shadow enhancers and find that the duplication of enhancers can decrease noise and increase fidelity, although at the metabolic cost of increased RNA production. A saturation mechanism for enhancer interactions similarly improves on both of these metrics. Taken together, this work highlights that shadow enhancer systems may exist for several reasons: genetic drift or the tuning of key functions of enhancers, including transcription fidelity, noise and output.

**Data Availability Statement:** All calculations and simulations were done using MATLAB 2016b under GCC C/C++ 4.9 in conjunction with the CERENA toolbox. The code is available at https://

## Author summary

During development, cells assume different fates based upon signals, including transcription factor proteins that bind to regions of the DNA called enhancers. Enhancers can interact with promoters to control the transcription of a target gene. Many developmental genes have multiple, seemingly redundant enhancers called *shadow enhancers*.

When each separate enhancer is bound by distinct transcription factors, shadow enhancers can drive less noisy gene expression than single enhancers. This allows for the buffering of perturbations in the transcription factor inputs. However, under this premise,

github.com/WunderlichLab/
TheoreticalEnhancerModels.git.

**Funding:** This research was partially supported by
NSF-Simons Center grant DMS1763272, Simons
Foundation grant 594598 to GE and ZW, NSF grant
DMS1616233 to GE, NIH grant R01HD095246 to
ZW, and the UC President's Dissertation Year
Fellowship to AF. The funders had no role in study
design, data collection and analysis, decision to
publish, or preparation of the manuscript.

**Competing interests:** The authors have declared
that no competing interests exist.

a single large enhancer bound by distinct transcription factors should also be capable of
buffering perturbations. Why then are shadow enhancers so prevalent?

Fletcher et al. developed computational models of enhancer-mediated transcription
that vary in the numbers of enhancers and transcription factor binding sites. They ana-
lyzed transcriptional properties in systems with and without shadow enhancers. The mod-
els revealed that shadow enhancers can provide a wider landscape of possible
transcriptional properties. This computational approach enabled a broader exploration of
shadow enhancer properties than is feasible experimentally and may guide future experi-
mentation. Given their prevalence in developmental gene regulation, investigation of
shadow enhancers may lead to a better understanding on the pathogenicity of certain
mutations found in developmental enhancers.

## 1 Introduction

Enhancers are non-coding regions of the genome that are bound by transcription factors
(TFs) and interact with the promoter to regulate transcription. Developmental genes are fre-
quently expressed in multiple tissues or time points and are regulated by multiple enhancers.
Traditionally, these enhancers were perceived as modular, with each driving a distinct portion
of a gene's spatiotemporal expression pattern, and together generating the entirety of a gene's
expression pattern. However, work in *Drosophila* revealed the presence of "shadow enhanc-
ers"—sets of two or more enhancers that control the same gene and drive identical or overlap-
ping expression patterns [1]. Similar enhancer groups have been identified in *C. elegans*, mice,
zebrafish, and humans [2–7]. Here, we aim to use theory and computational models to shed
light on the regulatory advantages of shadow enhancers, as well as under what conditions a
single enhancer and a set of shadow enhancers are interchangeable.

On the surface, shadow enhancers appear redundant—they drive overlapping expression
patterns and can often be knocked out without meaningfully affecting phenotype. However, in
multiple loci in both flies and mice, shadow enhancers are essential for driving normal devel-
opment under conditions of stress [8–11]. For instance, under high temperatures but not ideal
temperatures, deletion of a shadow enhancer for the *Drosophila* gene *snail* leads to abnormal
development of the *Drosophila* embryo. The earliest work describing "shadow enhancers" des-
ignated the enhancer farther away from the promoter as the shadow enhancer and the pro-
moter-proximal enhancer as the "primary enhancer." For the purposes of this work, we use
the more recently formulated definition of shadow enhancers, which refers to the entire group
of enhancers regulating the same target gene over space and time [12].

Progress in methods for identifying enhancers genome-wide has revealed the pervasiveness
of shadow enhancers in developmental gene loci. A study revealed that a majority of genes
involved in *Drosophila* muscle development are controlled by sets of three or more shadow
enhancers, and large-scale data analysis on mouse tissue samples from genomic databases
showed widespread shadow enhancer activity [11, 13, 14]. Meanwhile, in humans, assays
involving enhancer-derived RNAs suggested that approximately 80% of examined genes are
under the regulation of two or more shadow enhancers [15]. Overall, these results and others
suggest that most developmental genes in multi-cellular organisms are regulated by sets of
shadow enhancers.

In addition to ensuring proper development in stressful conditions, shadow enhancers
interact in multiple ways to fine-tune gene expression. For example, shadow enhancers might
be assumed to behave additively—the sum of their individual mRNA outputs equals that of

their combined output. However some shadow enhancers display subadditive or superadditive interactions, in which the combined activity of both enhancers is either less or more than the sum of their individual contributions, e.g. [16, 17]. Shadow enhancers can also repress one another, creating a composite gene expression pattern that is weaker or more restricted that either enhancer produces alone [18, 19]. Lastly, by binding distinct sets of input TFs, shadow enhancers can collectively buffer temporal noise in TF levels, yielding more consistent gene expression as a function of time [20].

Many of the mechanisms, e.g. synergy and repression, that are observed in shadow enhancers may also be achieved in single enhancers. This observation leads to the core question of this work: which properties are specific to shadow enhancers and not possible in single enhancer regulation? To do so, we focus on two important functions of enhancers. The first is their ability to faithfully translate an upstream signal, like a difference in TF concentration, into downstream expression output. That is, the enhancer should be capable of producing different expression levels in response to different TF concentrations. This feature is defined here as *fidelity*, and it is needed to generate output expression patterns that determine developmental cell fate in response to upstream signals. The second is their ability to buffer stochastic noise either from random fluctuations in the upstream signals or from internal enhancer dynamics. This feature is needed to buffer the noise that inevitably arises from molecular interactions to drive consistent expression patterns. For this purpose, we construct theoretical models of enhancer systems with different properties and analyze the resulting dynamics. In addition to exploring a larger number of configurations and parameter sets than is experimentally practical, we expect this approach to elucidate the selection pressures that can shape the creation of shadow enhancers and opportunities for transcriptional modulation that appear in the presence of shadow enhancers.

Previous work on theoretical models of shadow enhancers has proven to be fruitful in predicting and understanding the behavior of these systems. For example, a simple model of *hunchback* and *knirps* shadow enhancers by Bothma et al. [16] was used to show that frequent interactions among enhancers may lead to competition for promoter access and thus give rise to subadditive behavior. Likewise, a model of *hunchback* regulation of the even-skipped enhancers by Staller et al. [21] suggested that two different enhancers can recreate the same expression patterns with distinct regulatory logic. Grah et al. [22] studied a model related to the Monod-Wyman-Changeux hemoglobin system [23] in the context of enhancer regulation and considered several performance metrics. The work by Nousiainen et al. presented a computational framework for identifying model families that can predict enhancer activation dynamics in a mechanistic fashion [24]. In this work, we focused our approach on a set of minimalist reaction network models in which each of the reactions is stochastic, and the parameters were derived from previous transcriptional data of *Kruppel* enhancers [20]. Hence, we expect that this fully stochastic approach may capture nuances in the relationship between TF fluctuations and transcription.

In section 2.1, we describe a reaction network model of the *Kruppel* gene enhancer system. This is a minimal model developed and validated in Waymack et al. [20] that recapitulates the dynamics of enhancer-mediated transcription. We then describe how to use this model to approximate mean transcriptional output and derive the corresponding noise and fidelity. Section 2.2 describes our approach to generate similar models that differ in their numbers of enhancers and TF binding sites. By simulating these models, we can compare the effects of distinct shadow enhancer systems on transcriptional noise and fidelity. Since shadow enhancers can behave sub- or superadditively, we incorporate this behavior into our models in Section 2.3 and once again perform simulations to determine how the noise and fidelity are affected. In Section 2.4 we consider the concepts of transcriptional synergy and saturation. Finally, in

Section 2.5, we use our modeling framework to compare the changes in noise and fidelity resulting from an enhancer duplication and an enhancer splitting. We consider these events to be potential mechanisms for the origin of shadow enhancers and study them in this context.

# 2 Results

## 2.1 The *Kruppel* enhancer model

As a case study for our work, we use *Kruppel*, a gene required for early embryonic patterning in *Drosophila*. Around two hours post-fertilization, *Kruppel* is expressed in a stripe around the middle of the embryo, and this expression pattern is generated by a pair of shadow enhancers (Fig 1A). A minimal version of the *Kruppel* enhancer system was described in Waymack et al. [20] using the model in Fig 1B. Here, *A* corresponds to the proximal enhancer closest to the promoter, while *B* corresponds to the other, distal enhancer. The subscripts for *A* and *B* alternate between 0 and 1 to denote whether there is a TF bound to them at a given time. In this model, it is assumed that each enhancer interacts with the promoter immediately after a TF binds to it. Waymack, et al. measured the dynamic expression output of *Kruppel's* shadow enhancers individually and together. Using these data, we estimated the model parameters either directly or by using simulated annealing, which systematically simulates the model over several parameter sets until the difference between the model output and experimental data falls within an acceptable tolerance (Table 1.) To recapitulate our previously-published

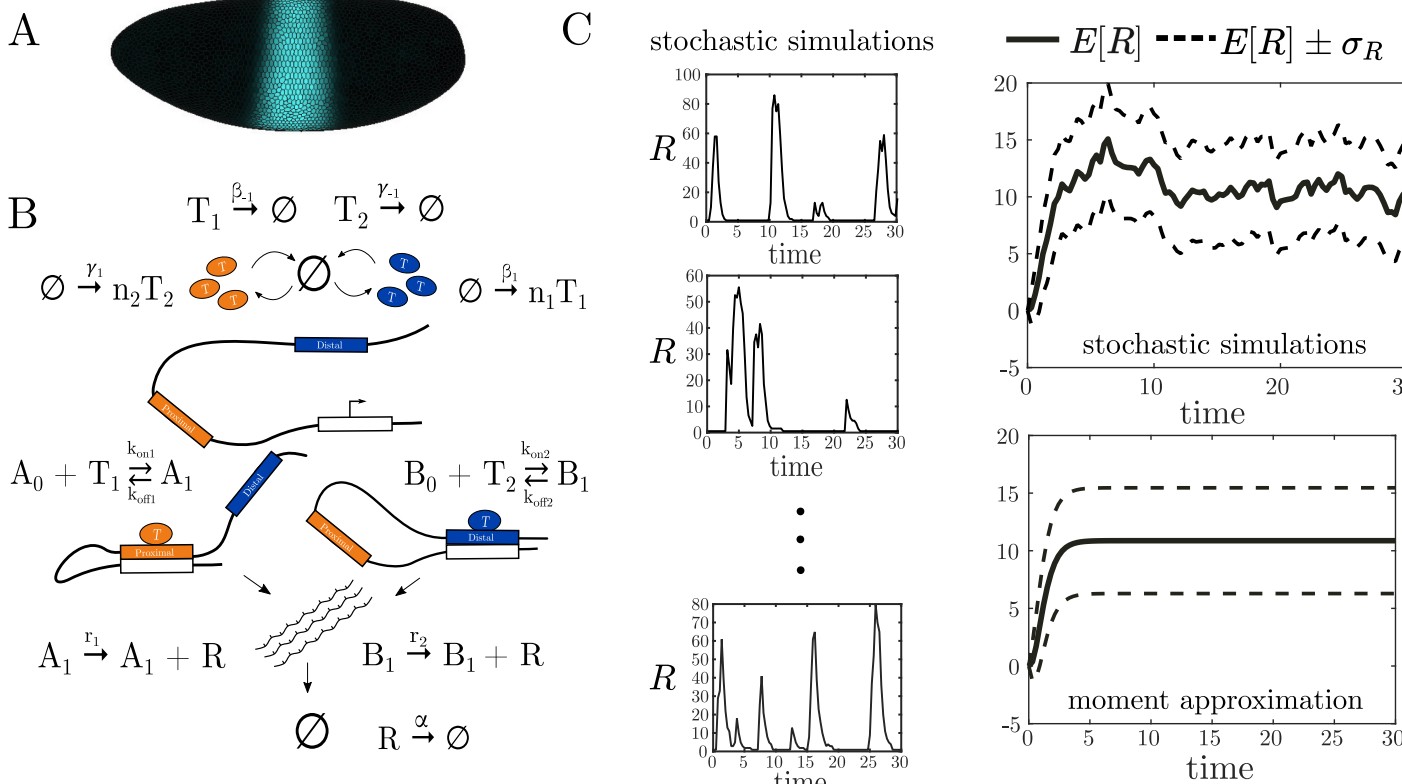

**Fig 1. Simulation of enhancer models and calculation of transcriptional noise and fidelity.** (A) A computational representation of the *Drosophila* embryo showing the region of *Kruppel* expression [20]. (B) Cartoon depicting a reaction network model of *Kruppel* shadow enhancers [20]. (C) Sample stochastic traces of mRNA from simulations of the model in (B) and their average over time $E[R]$ which estimates the mean mRNA concentration. The values of $E[R]$ and the standard deviation $\sigma_R$ can also be approximated by moment closure techniques and be used to estimate the transcriptional noise and fidelity of the modeled enhancer system.

**Table 1. Parameter values that were fitted to *Kruppel* expression data.**

| | Parameter Values |
|---|---|
| $\beta_1$ | 0.33 |
| $\beta_{-1}$ | 2.7 |
| $\gamma_1$ | 0.29 |
| $\gamma_{-1}$ | 3.9 |
| $k_{\text{off}_1}$ | 1.8 |
| $k_{\text{on}_1}$ | 0.36 |
| $k_{\text{off}_2}$ | 1.5 |
| $k_{\text{on}_2}$ | 0.19 |
| $\alpha$ | 1.96 |
| $r_1$ | 120 |
| $r_2$ | 140 |
| $n_1$ | 4 |
| $n_2$ | 12 |

experimental data, we stochastically simulated our model to yield the number of mRNA transcripts over time. Using the parameters in Table 1, simulated mRNA bursts resemble those observed during live transcription in *Drosophila* embryos (Fig 1C) and properties such as the size and duration of bursts were consistent with those from *Kruppel* experimental data [20]. Based on this work, the model shown in Fig 1B is used as a description of shadow enhancer dynamics.

One disadvantage of using stochastic simulations to estimate mean mRNA transcription is the significant computational cost. To do this more efficiently, we use moment closure methods that create ordinary differential equations (ODEs) describing the first and second moments of the chemical species [25, 26]. The solution to the ODE for the first moment approximates the limit to infinity of the mean mRNA denoted by $E[R]$. For example, the top right of Fig 1C shows $E[R]$ and the standard deviation $\sigma_R$ derived from a finite number of stochastic traces while the plot below shows the moment closure approximation of $E[R]$ and $\sigma_R$. Moment closure methods are popular tools to estimate measures of stochastic chemical reaction networks, but their reliability varies on a case by case basis. See [27, 28] for a comparison of different methods, and [29] for a set of sufficient conditions in which various approximations may hold.

Using the moment closure technique, we can now efficiently estimate the transcriptional fidelity and noise of our modeled enhancer system. To capture the transcriptional fidelity, we used the correlation between the TF values and the corresponding levels of gene expression. The correlation $Corr(T_1, R)$ quantifies the degree to which the first TF, $T_1$ and mRNA concentrations rise or fall together. It aims to describe whether a large (or small) $T_1$ input has a tendency to lead to a large (or small) mRNA output concentration. To study the transcriptional noise, we used the coefficient of variation (CV) by normalizing the standard deviation of mRNA expression by its mean. We have that

$$CV = \frac{\sigma_R}{E[R]},$$

while

$$Corr(T_1, R) = \frac{cov(T_1, R)}{\sigma_{T_1}\sigma_R}.$$

Notice that both of these quantities can be directly calculated from the first and second moments of the different species, which are in turn estimated using moment closure methods.

## 2.2 Models with varying number of enhancers and total binding sites

Using the *Kruppel* model as a starting point, we first wanted to explore if transcriptional fidelity and noise are dependent on TF binding sites being arranged into two different enhancers. Our previous experimental and computational work found that having two enhancers with distinct TF binding sites—$T_1$ in the proximal enhancer and $T_2$ in the distal enhancer—drove lower expression noise than two enhancers with identical TF binding sites. From this work, however, it was not clear whether the TF binding sites had to be split among two enhancers, or whether a single enhancer with sites for both $T_1$ and $T_2$ could achieve the same noise reduction. It also was not clear if there might be a trade-off between noise and fidelity.

To explore these questions, we constructed 48 models with up to four enhancers and four total binding sites, as shown in Fig 2A. Each enhancer has a given number of binding sites for $T_1$ and $T_2$, and every diagram in this figure corresponds to a set of chemical reactions. For an

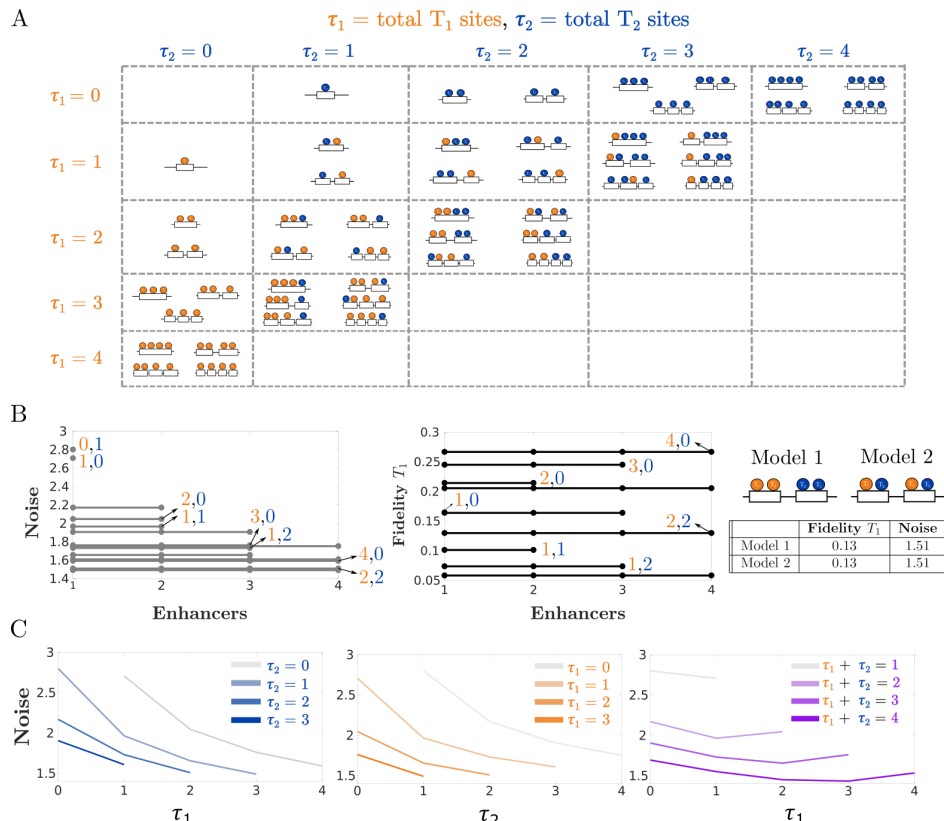

**Fig 2. Under additive assumptions, transcriptional fidelity and noise are independent on enhancer number but depend on TF binding site number.** (A) Different enhancer models used in the simulations. Each model has different total binding sites for $T_1$, total binding sites for $T_2$, distribution of binding sites, and number of enhancers. (B) Simulations for the models in (A) show that fidelity and noise are independent of the number of enhancers and the distribution of binding sites. The noise broadly decreases as a function of total TF binding sites, while fidelity with respect to $T_1$ increases with the number of $T_1$ binding sites. The table on the right shows the fidelity and noise values for two different configurations of TF binding sites among two enhancers. (C) Noise calculated as functions of the total binding sites for $T_1$ or $T_2$. As the total number of binding sites increases, the noise generally decreases.

example of the specific networks associated with two of these diagrams, see S1 Fig in the Supplementary Material. We then input each network into the CERENA software (ChEmical REaction Network Analyzer) [26], which allows us to calculate the ODE equations for the mean and other moments of all chemical species in the network. CERENA uses an array of approximation methods that allow for a streamlined implementation of these ODE equations, in particular several methods of moment closure. The use of this software is significantly faster as compared with many repeated runs of the Gillespie algorithm [30]. We calculated ODEs up to the second moments in order to calculate the variance, and we used the method of zero cumulants for moment closure [31]. The results were found to be generally consistent with simulations of individual models carried out with the more standard Gillespie algorithm (Fig 1C, right).

To create the reaction equations for all our configurations of interest, we needed to decide how to assign the model parameters in a way that made the comparisons meaningful. Most of the parameters, including the $k_{on}$ and $k_{off}$ rates, were assigned the same values found to be consistent with experimental data in Waymack et al. [20] and used in Fig 1C. See Table 1 for the values of the different parameters in the model. The rates of transcription for each enhancer state are initially defined using the following additive scheme. Any enhancer with $n$ bound $T_1$ and $m$ bound $T_2$ will produce mRNA at a rate $nr_1 + mr_2$ where $r_1$ and $r_2$ are the rates of transcription of the *Kruppel* model in Fig 1B. Similarly, if two or more shadow enhancers are bound to the promoter, the overall mRNA transcription rate is the sum of the transcription rates for each enhancer. This way, we have an additive scenario where a single enhancer system with four fully bound sites of $T_1$ will produce mRNA at the same rate as four independent enhancers that are each bound by a single $T_1$.

We calculated the transcriptional noise and fidelity for all the models in Fig 2A to discern the effect of different configurations of binding sites and enhancers. The results are shown in Fig 2B, left (S2 Fig shows the results for fidelity with respect to $T_2$). As long as the number of binding sites for $T_1$ and $T_2$ remain constant, the noise does not change when altering the number of enhancers. Moreover, even after fixing the number of enhancers, the actual distribution of TF binding sites among the enhancers appears to have no effect on transcriptional noise or fidelity (Fig 2B, right). The fidelity to $T_1$ is also insensitive to the number of enhancers, but increases for increasing numbers of $T_1$ binding sites, and it decreases when additional $T_2$ binding sites are present (Fig 2B, middle). One can conclude that, in this additive regime, there may be negligible selection pressures on the number of enhancers regulating a gene and the distribution of TF binding sites among these enhancers.

Though the noise does not depend on enhancer number, it does vary with binding site number. The plots in Fig 2C show that enhancers with more binding sites generally lead to lower transcriptional noise. This may be because as the number of binding sites increases, there are more inputs in the system, which can serve to average each other out. Each binding site can be thought of as akin to a coin toss. When $n$ independent coin tosses are made, the expected number of heads grows linearly but its standard deviation grows sublinearly, leading to an overall decrease in the noise.

Not only does transcriptional noise depend on binding site number, but it is also dependent on binding site identity. The production reactions of each of the two transcription factors are independent of each other. Shifting the total number of binding sites $\tau_1 + \tau_2$ shows non-monotonic changes in noise (rightmost plot in Fig 2C). The leftmost data points at $\tau_1 = 0$ correspond to enhancers that are bound exclusively by $T_2$ while the rightmost points at $\tau_1 = 4$ correspond to enhancers that are bound exclusively by $T_1$. As $T_1$ sites increase, the ratio between $T_1$ and $T_2$ binding sites becomes more even and the noise decreases. However, once enhancers adopt too many $T_1$ sites at the expense of $T_2$ sites, noise begins to increase. This agrees with previous

observations that having a single kind of TF binding site leads to higher noise in transcription than systems bound by multiple kinds of TF binding sites [20]. In summary, these models demonstrate that when enhancer output combines in an additive fashion, the fidelity does not depend on the number of enhancers. Noise is sensitive to the number and identity of binding sites, but not to their arrangement among one or more enhancers. Thus, the prevalence of shadow enhancers under additive assumptions might be the result of genome dynamics and genetic drift, as opposed to selection.

To verify the accuracy of the chosen moment closure method used for the simulations in this section and sections below, we carried out a comparison with a higher order zero-cumulant method as well as other methods, see the Supplementary Material (S5 Fig). While there is some variability between methods, the noise and fidelity measurements are broadly preserved.

## 2.3 Subadditivity and superadditivity

The results in the previous section showed that the number of additive enhancers does not affect transcriptional noise and fidelity. In turn, the number of additive enhancers may be under minimal selection pressure and instead originate through stochastic processes such as genetic drift. However, shadow enhancers have been observed to behave subadditively and superadditively—their combined activity either results in a lesser or greater amount of gene expression than the sum of their independent contributions. To investigate whether different levels of additivity could result in different fidelity and noise properties, we modified our models in the previous section to recapitulate this effect.

To capture varying enhancer additivity in our models, we increased or decreased the binding rates $k_{on}$ and $k_{off}$ to approximate the biochemical mechanisms underlying varying levels of additivity. We chose this approach to implementing sub- or superadditivity because it was consistent with previous experimental data [20]. Using this approach, any enhancer model can be made subadditive by decreasing $k_{on}$ and increasing $k_{off}$. Similarly, enhancer models can be made superadditive by increasing the values of $k_{on}$ and decreasing the values of $k_{off}$. In addition, in a later section we will briefly consider the cases where the polymerase loading rates are saturated by a single enhancer or remain null until all enhancers bound to the promoter.

For simplicity, we used a linear relation between the number of enhancers and the binding and unbinding parameters. Specifically, in the subadditive case the "new" binding rate is defined as $k_{on}^{new} = k_{on} - d_1 n$, and similarly $k_{off}^{new} = k_{off} + d_2 n$ (Figs 3A and 4A). We selected values of $d_1$ and $d_2$ to broadly explore the effect of subadditivity while still allowing the moment closure method to accurately estimate noise and fidelity.

Unlike the additive case, when enhancers interact subadditively, their numbers affect the noise (Fig 3B and S2 Fig). The fidelity is however mostly unaffected by the number of enhancers. Similar to the additive case, binding site numbers in subadditive enhancers can affect the fidelity as well as the transcriptional noise. Generally, given a fixed number of enhancers, adding TF binding sites will reduce the noise, while adding $T_1$ binding sites will increase the fidelity. Moreover, subadditive systems with binding sites for both $T_1$ and $T_2$ are less noisy than systems with a single kind of binding site (Fig 3C). The distribution of binding sites among enhancers has no effect on the fidelity or the noise. In sum, the subadditive models are distinct from the additive models in that they show that increasing enhancers numbers increases transcriptional noise while preserving fidelity. However, the relationship between the number and type of binding sites and transcriptional noise are similar between the additive and subadditive cases.

Superadditive shadow enhancers have also been observed, so to explore their properties, we linearly increased $k_{on}$ and decreased $k_{off}$. Systems with superadditive enhancers show

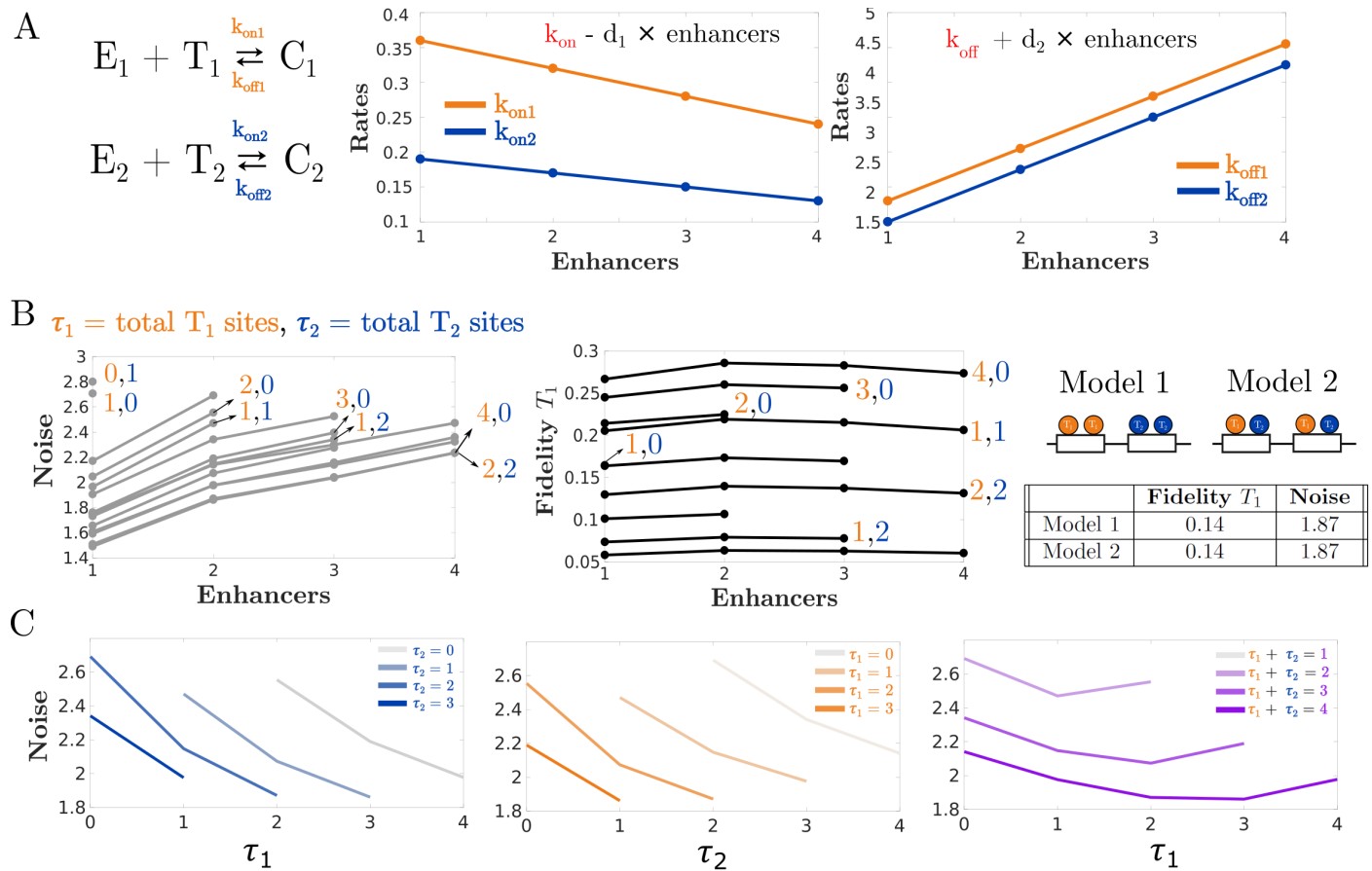

**Fig 3. In subadditive enhancers, noise increases with enhancer number but fidelity is broadly unchanged.** (A) Subadditivity is implemented in our model by linearly decreasing $k_{on}$ rates and linearly increasing $k_{off}$ rates. In this case $d_1$, the rate of decrease for $k_{on}$, was chosen to be 0.04 for $T_1$ and 0.02 for $T_2$. Meanwhile $d_2$, the rate of increase for $k_{off}$, was chosen to be 0.75 for both $T_1$ and $T_2$. (B) Systems with more subadditive enhancers tend to exhibit higher noise, while the fidelity is broadly independent of enhancer number. Noise and fidelity are also independent of binding site distribution but vary with respect to the number of binding sites. (C) Plots showing the relationship between binding site numbers and transcriptional noise for two subadditive enhancers.

decreasing transcriptional noise as well as decreased fidelity with increasing numbers of enhancers. In this way, superadditive systems present a trade off between noise and fidelity—increasing enhancer numbers improve noise at the expense of fidelity (Fig 4B).

Once again, binding site distributions in superadditive enhancers do not affect the fidelity or the noise. However the total number of binding sites does affect the noise and fidelity as in the previous two cases (Fig 4C and S4 Fig).

## 2.4 Saturation and synergy

Another way that shadow enhancers can interact is through transcriptional saturation or synergy. In a system with $n$ shadow enhancers, the system presents *saturation* if the transcription rate $r$ is the same if at least one enhancer is bound to the promoter. On the contrary, the system presents *synergy* if the rate of transcription is 0 unless all $n$ enhancers are bound to the promoter. We show the results of such systems in Fig 5. For simplicity, we use a single type of transcription factor and we vary the number of enhancers.

In the saturation case, increasing the number of enhancers will reduce the noise as well as increase fidelity. Both noise and fidelity change about two-fold as the system increases from

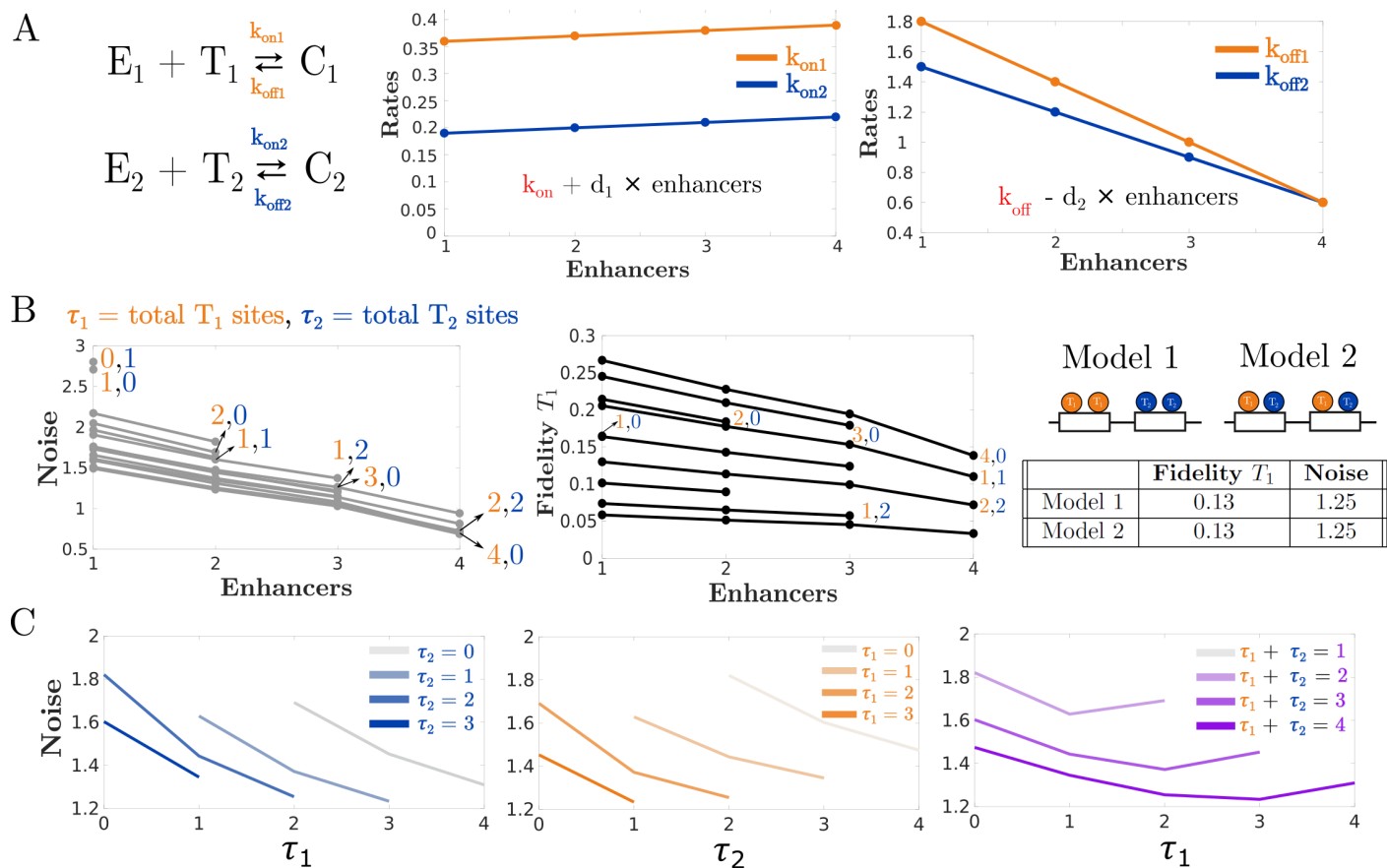

**Fig 4. In superadditive networks, more enhancers decrease noise and fidelity.** (A) Superadditivity is implemented in our model by linearly increasing $k_{on}$ rates and linearly decreasing $k_{off}$ rates. In this case $d_2$, the rate of decrease for $k_{off}$, was chosen to be 0.4 for $T_1$ and 0.3 for $T_2$. Meanwhile $d_1$, the rate of increase for $k_{on}$, was chosen to be 0.01 for both $T_1$ and $T_2$ (B) Unlike in the subadditive case, enhancer numbers decrease transcriptional fidelity and also decrease noise. The distribution of binding sites does not affect either the noise or the fidelity all else being constant. (C) Plots showing the relationship between binding site numbers and transcriptional noise for two superadditive enhancers. Increasing binding site numbers leads to less noise in gene expression.

one to four enhancers. In the synergy case, the noise increases, and the fidelity decreases, as the number of enhancers increases. A significant noise level of almost 30 was reached for four enhancers. In this sense, the saturation model can be considered the more efficient one in terms of the studied metrics.

## 2.5 Shadow enhancer duplication

While not explicitly stated, the models in Fig 2A correspond to a collection of single enhancers that split into multiple enhancers. Analyzing our results through this lens would suggest that the splitting of an enhancer into subadditive or superadditive enhancers could occur based on a need to modulate noise, fidelity, or absolute levels of gene expression. For additive enhancers, such a split would not face any adverse selection pressures based on the absence of changes to the mRNA output. However, the mechanisms by which shadow enhancers come to being remain unclear and the splitting of an enhancer might not be the only viable route to create shadow enhancers [12]. Hence, to contrast our results with a different mechanism, we apply our analysis to shadow enhancer systems that arise by enhancer duplication.

First, we generated models that are repeated duplications of single enhancer models. Cartoons depicting these models are shown in Fig 6A. In some cases, where each enhancer had

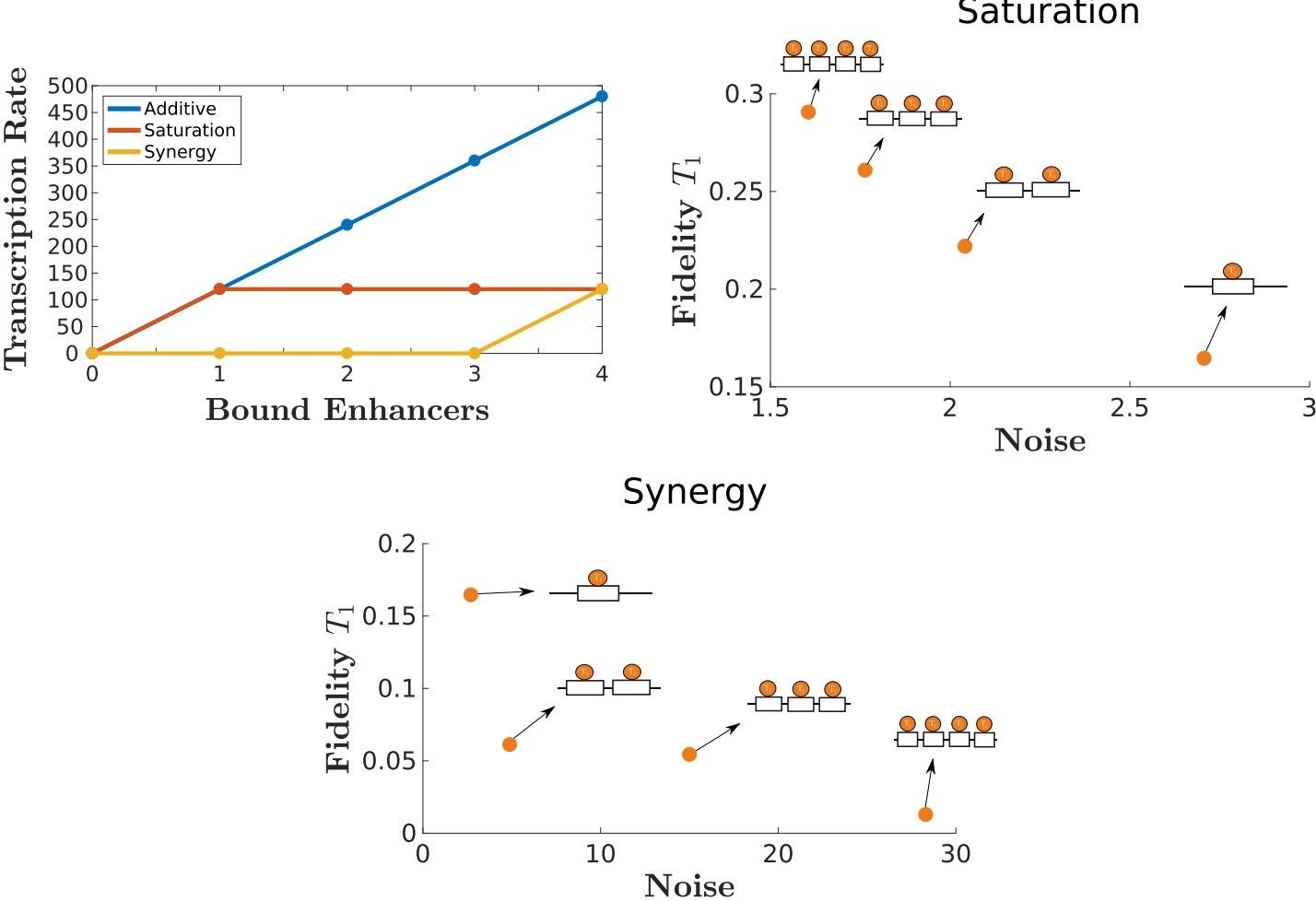

**Fig 5. Saturating and synergistic enhancer interactions lead to different trends in noise and fidelity.** A saturated system yields mRNA at the same rate for any positive number of enhancers bound. On the other hand, a synergistic system becomes active only when all enhancers are bound to the promoter. The resulting plots of fidelity and noise corresponding to these systems show inverse relationships between noise and fidelity. In the saturating regime, low noise and high fidelity are achieved with higher enhancer numbers, while in the synergistic regime, low noise and high fidelity occur with lower enhancer numbers.

multiple binding sites, we had to limit the number of enhancers simulated due to their large number of reactions and the associated computational costs. Given the experimental observations of subadditivity with enhancer duplication, we focused our analysis on the duplication of subadditive enhancers and compared it with the alternative of splitting subadditive enhancers [17, 20].

Unlike splitting, enhancer duplication not only increases the number of enhancers but also the number of binding sites (Fig 6B). For this reason, the number of binding sites for each model in the plots of Fig 6C is shown as a product of the binding sites per enhancer times the number of enhancers. These plots show that the duplication of subadditive enhancers leads to a slight decrease in the noise while increasing transcriptional fidelity. The effect of duplication on transcriptional noise is consistent with the experimental measurements of the *Kruppel* enhancers [20].

Ultimately, enhancer duplication shows the potential for increasing transcriptional fidelity while simultaneously reducing expression noise, albeit with the additional metabolic cost of increasing RNA output. The duplication of subadditive enhancers presents similar trade-offs

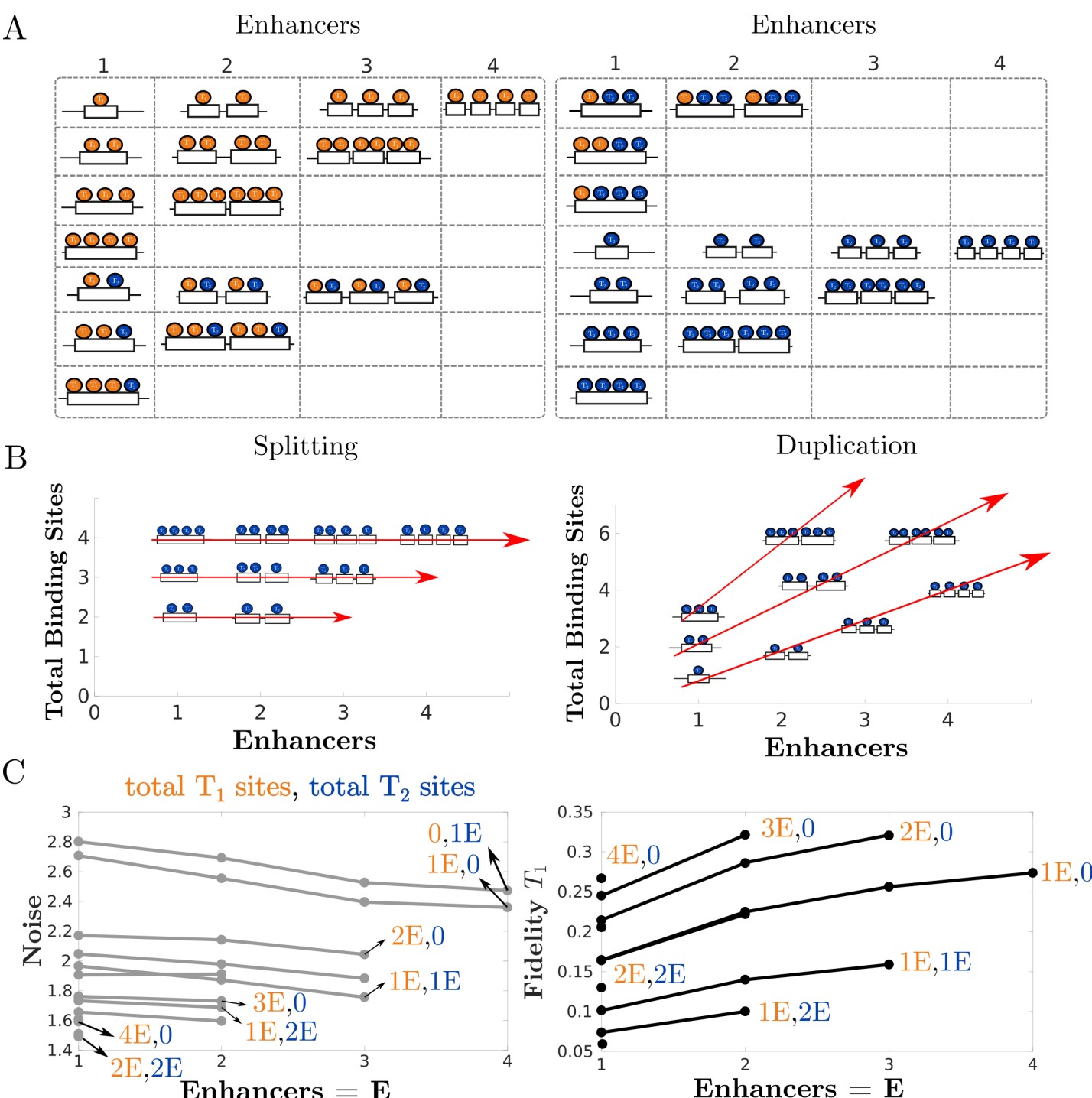

**Fig 6. Duplication of subadditive enhancers can increase transcriptional fidelity and reduce noise.** (A) Single enhancer models and those that result from repeated enhancer duplications. (B) Plots showing the relationship between total binding sites and enhancers for the case of enhancer splitting and enhancer duplication. Splitting does not affect total binding site numbers while it increases enhancer numbers. On the other hand, duplication increases both enhancer numbers and total binding sites at different rates. (C) Plots depicting transcriptional fidelity and noise for subadditive versions of the models in (A). Enhancer duplications increase transcriptional fidelity while the noise generally decreases.

from those observed in the superadditive splitting scenario and expands the possibilities of transcriptional modulation from those available for single enhancer systems.

## 3 Discussion

In this work, we sought to determine the effect of varying the number of shadow enhancers as well as the nature of their interactions, and to understand whether a single enhancer can recapitulate their dynamical behavior. To do so, we simulated models with differing numbers of enhancers and TF binding sites and calculated the transcriptional noise and fidelity for each model. Sufficiently high fidelity is required for a gene's expression to meaningfully reflect changes in upstream signals intended to shape a cell's fate. On the other hand, sufficiently low noise is needed for transcription to convey consistent signals of expression in the face of unavoidable molecular fluctuations. Consequently, a balance is needed between these properties in order to properly pattern a developing organism.

We began by considering additive enhancers—enhancers which have a combined transcriptional rate equal to the sum of their individual contributions. Our models revealed that the number of additive enhancers has no effect on transcriptional noise and fidelity.

Therefore, in the case of additive shadow enhancers, there would seem to be no particular pressure to have multiple shadow enhancers, as opposed to one large enhancer. How might we explain the preponderance of shadow enhancers in this case? Here, the dynamics of genome evolution may be at play. If we imagine that a genome contains a single, large enhancer, there are processes, like transposable element insertions [32, 33] or DNA polymerase slippage [34, 35] that may split this enhancer into shadow enhancers. If these shadow enhancers can control gene expression similarly to the ancestral single enhancer, as our model suggests, there would be no particular pressure to remove the intervening sequence. In fact, it may be entropically more favorable to split an enhancer into shadow enhancers that to merge shadow enhancers into a single enhancer, given that many more splitting events could lead to two functional enhancers, while deletion or excision events would have to be a great deal more specific to avoid removing functional enhancer sequence.

Many experimental studies have shown that shadow enhancers can interact in a wide range of manners—sub- and superadditively, as well as repressively [16–19, 36–38]. We therefore modified our models to recapitulate this behavior. The resulting simulations revealed that increasing numbers of subadditive enhancers corresponded to lower levels of transcriptional noise while broadly maintaining the levels of fidelity. On the other hand, increasing numbers of superadditive enhancers led to a trade off between a desirable decrease in noise with an undesirable decrease in fidelity. We also found in the transcriptional saturation model that including more enhancers can lead to a significant increase in fidelity with a decrease in the noise, a potentially strong alternative to improve on both metrics. Notice that increasing the number of enhancers increases the metabolic cost of the system [39]. In general, when shadow enhancers interact in more intricate ways, there is a complex landscape of transcriptional noise, fidelity, and output that selection may act upon to determine the number of enhancers controlling a gene.

Though there are many hypotheses about the mechanisms that drive enhancers to act sub or superadditively, like enhancer competition for the promoter or synergy between recruited TFs and co-factors, we are not yet able to predict how two or more enhancers will interact [16, 40]. Superadditivity and subadditivity may be structural properties that may not be tunable, but instead are the result of existing constraints for each individual system. Thus there are likely many mechanisms by which enhancers become non-additive and other evolutionary trade-offs beyond transcriptional noise and fidelity that elude our simplified models.

We also wished to examine the impacts on noise and fidelity of different mechanisms that could result in the creation of shadow enhancers [12, 41]. In particular, we wanted to contrast the case of an enhancer that splits into multiple enhancers and an enhancer that duplicates itself. To study enhancer duplication, we adapted our models to this scenario and noted that repeated duplications of subadditive enhancers induced lower transcriptional noise alongside increases in transcriptional fidelity. It remains unclear whether splitting or duplication of genomic regions are common mechanisms for the origin of shadow enhancers and whether they presents additional trade-offs with respect to genome size. Our analysis was also limited in that it assumes exact duplications of each enhancer, which is inconsistent with the stochasticity typically involved in genome duplication [42].

Naturally, our approach was also limited by the constraints we imposed on our models. The linear approach to modeling sub- and superadditivity has a limitation, in that given a sufficiently high number of enhancers, the $k_{on}$ or $k_{off}$ rates can eventually become negative. We implemented this approach for simplicity, but it could also be done, e.g. with exponentially rather than linearly decreasing values to prevent this effect. While we use the CERENA package to estimate the moments of these systems, other methods such as the Linear Mapping Approximation [43] are available and could potentially be used in such networks in the future. This method in particular can be used for second order reactions in which a protein binds a gene, and it does not rely on assuming a particular distribution for the stochastic variables in the system. In S5 Fig, we present a comparison of the second order zero-cumulant method used for the moment closure with other methods, showing that our methodology is broadly compatible with other forms of moment closure. We also show calculations of the Fano factor for this system.

Throughout this work, we used parameters that were fitted to experimental data derived from studying the *Kruppel* enhancer system. This parameter space, however, may not be in accordance with the dynamics of other enhancer systems found in *Drosophila* and other organisms. To address this, in S6 Fig we carry out a basic random variation of the parameter set over several orders of magnitude, showing that the results in our main figures are broadly consistent in these other simulations. Regarding S6 Fig, notice that despite the general trends, it is still possible to find parameter sets for which the trends do not hold.

Our modeling suggests experiments that could prove useful in further exploring the mechanisms of shadow enhancer function. For instance, one could imagine constructing a synthetic system within an embryo to directly test the impact of varying enhancer numbers and TF binding site distribution on the associated noise and fidelity of the transcriptional output [44, 45]. In addition, a bioinformatic analysis comparing enhancer DNA sequences in different organisms could be used to determine the relative prevalence of enhancer splitting, duplication, and other ways of generating new shadow enhancers.

Overall, this work shows that different strategies for shadow enhancer interaction present a variety of trade-offs. In the additive scenario, increasing the number of enhancers appears to face no transcriptional trade-offs or advantages. In the subadditive scenario, additional enhancers increase noise but might be tolerated if additional mRNA output is needed. On the other hand, superadditive enhancers could supersede a single enhancer decreased noise is needed more than fidelity. Shadow enhancers that combine in a saturating manner can give rise to both increased fidelity and decreased noise with increasing enhancer numbers. We also showed that repeated enhancer splittings lead to distinct outcomes in noise and fidelity from repeated duplications. Ultimately, the preponderance of shadow enhancers may be due to a combination of genetic drift and to the variety of transcriptional modulation strategies possible with multiple, but not single enhancers.

## 4 Methods

### 4.1 Description of enhancer models and parameters

The model of *Kruppel* gene enhancers in Fig 1B is described by the following chemical reaction network.

$$T_1 + A_0 \underset{k_{\mathrm{off}_1}}{\overset{k_{\mathrm{on}_1}}{\rightleftharpoons}} A_1 \quad \overset{r_1}{\rightarrow} \quad A_1 + R$$

$$T_2 + B_0 \underset{k_{\mathrm{off}_2}}{\overset{k_{\mathrm{on}_2}}{\rightleftharpoons}} B_1 \quad \overset{r_2}{\rightarrow} \quad B_1 + R$$

$$\emptyset \overset{\beta_1}{\rightarrow} n_1 T_1$$

$$\emptyset \overset{\gamma_1}{\rightarrow} n_2 T_2$$

$$T_1 \overset{\beta_{-1}}{\rightarrow} \emptyset$$

$$T_2 \overset{\gamma_{-1}}{\rightarrow} \emptyset$$

$$R \overset{\alpha}{\rightarrow} \emptyset.$$

Here, two enhancers denoted as *A* and *B* have a single binding site for their respective TFs $T_1$ and $T_2$. The subscripts for each enhancer reflect how many TFs are bound to them. For example, $A_0$ denotes the enhancer *A* with no TFs bound while $A_1$ denotes the same enhancer with a single TF bound. Our modeling framework assumes that a single TF bound to the enhancer is sufficient for an enhancer-promoter interaction. Once an enhancer is bound to the promoter, the rates of transcription correspond to a linear combination of the TFs bound to that enhancer. In this case, since *A* and *B* possess only one binding site, the transcriptional rates are only the single terms $r_1$ and $r_2$. The TFs $T_1$ and $T_2$ appear in clusters of sizes $n_1$ and $n_2$ and dissipate or degrade linearly at rates $\beta_{-1}$ and $\gamma_{-1}$. These particular properties of TFs were previously noted to be necessary for recapitulating *Kruppel* transcriptional data. Lastly, mRNA denoted by *R* degrades at a constant rate $\alpha$. Details about fitting the reaction rates to experimental data and further justification for the topology of this model can be found in Waymack et al. [20].

We expanded the reasoning above to construct models for any number of enhancers each with an arbitrary number of binding sites. In particular, take a set of *n* enhancers given by $A^{(1)}$, $A^{(2)}, \ldots, A^{(n)}$. We denote an enhancer $A_I^{(i)}$ where *I* is a vector composed of 0s or 1s with the $k^{\mathrm{th}}$ entry being a 1 if there is a TF bound to the $k^{\mathrm{th}}$ binding site of $A^{(i)}$ and 0 otherwise. Denote $e_j$ as the vector with 0s at all entries except for a 1 at the $j^{\mathrm{th}}$ entry and having the same number of entries as *I*. Any binding of a TF $T_m$ to the (empty) $j^{\mathrm{th}}$ binding site of $A_I^{(i)}$ is described by a reaction of the form

$$A_I^{(i)} + T_m \overset{k_{\mathrm{on}_m}}{\rightarrow} A_{I+e_j}^{(i)}$$

Similarly, an unbinding of $T_m$ from $A_I^{(i)}$ is described by the reaction

$$A_I^{(i)} \xrightarrow{k_{\text{off}_m}} A_{I-e_j}^{(i)} + T_m$$

Suppose the enhancer $A_I^{(i)}$ is bound by TFs $T_{l_1}, T_{l_2}, \ldots, T_{l_p}$ where the binding sites for an arbitrary TF $T_{l_c}$ are located between the entries $f_1^{(c)}$ and $f_2^{(c)}$ of $I$. Then any enhancer with subscript $I$ will initiate transcription at a rate equal to $\sum_{c=1}^{p} \sum_{d=f_1^{(c)}}^{f_2^{(c)}} I_d r_{l_c}$ where $I_d$ is the $d^{\text{th}}$ entry of $I$. Lastly, any TF $T_m$ appears in clusters of size $n_m$ as described by the reaction

$$\emptyset \rightarrow n_m T_m$$

and degrades at a linear rate. Two concrete examples of this procedure are shown in S1 Fig for enhancer systems that were included in Fig 2A.

## 4.2 Estimation of noise and fidelity

To estimate the noise in mRNA expression for all the models in Fig 2A, we approximated the moments for $R$ using zero cumulants closure with a second order truncation. In other words, we assumed that all cumulants of the specified output with order larger than 2 were negligible when calculating the moments. However, unlike the simple example shown above, our models may have several dozen reactions which can make the process of calculating the moments extremely laborious. Fortunately, the CERENA toolbox provides a suite of moment closure methods, including zero cumulants closure, that are conveniently arranged to take reaction networks as inputs [26]. We generated these input files for the 48 models in Fig 2A using a Python script that takes as inputs a number of enhancers and TF binding sites and generates the corresponding model file for CERENA. Then, using the moments that were calculated with CERENA, we derived the correlation and CV for each of the models. These measurements were plotted in Figs 2, 3, 4 and 6.

## 4.3 Implementation of sub and superadditivity

The implementation of varying additivity levels was done by modulation of the binding rates $k_{\text{on}}$ and $k_{\text{off}}$ as shown in Figs 3A and 4A. For example, to make the *Kruppel* model in Fig 2A subadditive, we would proceed as follows. Let $k_{\text{on}_1}^{(1)}$ and $k_{\text{off}_1}^{(1)}$ be the rates of binding and unbinding for $T_1$ to and from a single enhancer. Since the model in Fig 2A has two enhancers, we would set $k_{\text{on}_1}$ in this model equal to $k_{\text{on}_1}^{(1)} - 2d_1$ and $k_{\text{off}_1}$ to $k_{\text{off}_1}^{(1)} + 2d_2$ where $d_1$ and $d_2$ are positive constants. Then, repeat the same procedure for the binding and unbinding rates $T_2$ for another set of $d_1$ and $d_2$ values. This way, the TFs will bind to the enhancers less often leading to an overall decrease in mRNA production. This approach was found to be sufficient for consistently recapitulating the subadditivity of *Kruppel* enhancers observed in the work by Waymack et al [20]. Superadditivity, on the other hand, can be implemented for the model in Fig 2A in a similar way by having $k_{\text{on}_1}$ equal to $k_{\text{on}_1}^{(1)} + 2d_1$ and $k_{\text{off}_1}$ to $k_{\text{off}_1}^{(1)} - 2d_2$. The same procedure can also be applied to the binding rates of $T_2$.

In general, any model can be made subadditive according to the following scheme. Take once again the system with $n$ enhancers given by $A^{(1)}, A^{(2)}, \ldots, A^{(n)}$ as described above. Denote $k_{\text{on}_m}^{(1)}$ and $k_{\text{off}_m}^{(1)}$ be the rates of binding and unbinding for some TF $T_m$ to and from a single

enhancer. Then, set $k_{\mathrm{on}_m}$ in the reactions of the form

$$A_I^{(i)} + T_m \xrightarrow{k_{\mathrm{on}_m}} A_{I+e_j}^{(i)}$$

and $k_{\mathrm{off}_m}$ in the reactions of the form

$$A_I^{(i)} \xrightarrow{k_{\mathrm{off}_m}} A_{I-e_j}^{(i)} + T_m$$

equal to $k_{\mathrm{on}_1}^{(1)} - nd_1^{(m)}$ and $k_{\mathrm{off}_1}^{(1)} + nd_2^{(m)}$ respectively for some positive constants $d_1^{(m)}$ and $d_2^{(m)}$. Lastly, repeat these steps for all values of $m$. The same reasoning can be applied when designing superadditive systems but the signs in the equations of binding rate modulation need to be flipped, that is, use instead $k_{\mathrm{on}_1}^{(1)} + nd_1^{(m)}$ and $k_{\mathrm{off}_1}^{(1)} - nd_2^{(m)}$ for modifying the corresponding binding rates. These procedures were implemented for all the models in Fig 2A and the resulting transcriptional noise and fidelity for these modified models was plotted in Figs 3 and 4.

## Supporting information

**S1 Fig. Enhancer network structure.** Two different models and their corresponding reaction networks.
(EPS)

**S2 Fig. Transcriptional fidelities with respect to $T_2$ for enhancers that behave additively, subadditively, and superadditively.** The fidelity trends with respect to $T_2$ for all models in Fig 2A do not differ significantly from those corresponding to the fidelities with respect to $T_1$.
(EPS)

**S3 Fig. Fitted rates of mRNA transcription for single and duplicated models suggest that a single enhancer is sufficient to saturate polymerase loading rates.** The polymerase loading rates $r_1$ and $r_2$ were fitted for the 4 models shown above according to the same methodology described in Waymack et al. [20]. Parameter fittings were done directly on the raw mRNA transcriptional data of *Kruppel* and show minimal differences between the single enhancer models and their duplicated counterparts.
(EPS)

**S4 Fig. Increasing the binding site numbers in systems with three enhancers leads to decreases in noise.** Noise trends with respect to binding site numbers for systems with three enhancers that operate subadditively and superadditively. Higher binding site numbers lead to lower transcriptional noise.
(EPS)

**S5 Fig. (A) Comparison of different moment closure methods using the CERENA package**. From left to right, we use the zero cumulant methods of second and third order, mean field of second order, low dispersion of second and third order, and derivative matching of second order. **(B) Calculation of the Fano factor, defined as variance over mean, for the mRNA**. The graphs correspond to the calculations of the coefficient of variation in Figs 2B, 3B and 4B.
(EPS)

**S6 Fig. Additional parameter sets.** Four new sets of parameters were randomly chosen to recreate several graphs from previous figures. All parameters but $r$, $k_{\mathrm{on}}$ were randomized logarithmically from 0.1 to 100, while $r$ was randomized from 10 to 1000 and $k_{\mathrm{on}}$ was randomized from 0.1 to 10. The original parameter set is shown in red, while new parameter sets are shown

in blue, green, black, and purple. The figures expand on calculations presented in (A) Fig 2B for the additive case, (B) Fig 3B for the subadditive case, and (C) 4B for the superadditive case, as well as S2 Fig.
(EPS)

## Author Contributions

**Conceptualization:** Alvaro Fletcher, Zeba Wunderlich, German Enciso.

**Data curation:** Alvaro Fletcher.

**Formal analysis:** Alvaro Fletcher, Zeba Wunderlich, German Enciso.

**Funding acquisition:** Alvaro Fletcher, Zeba Wunderlich, German Enciso.

**Investigation:** Alvaro Fletcher.

**Methodology:** Alvaro Fletcher, Zeba Wunderlich, German Enciso.

**Project administration:** Zeba Wunderlich, German Enciso.

**Software:** Alvaro Fletcher.

**Supervision:** Zeba Wunderlich, German Enciso.

**Visualization:** Alvaro Fletcher, Zeba Wunderlich, German Enciso.

**Writing – original draft:** Alvaro Fletcher, Zeba Wunderlich, German Enciso.

**Writing – review & editing:** Alvaro Fletcher, Zeba Wunderlich, German Enciso.

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
