## [Decision Letter · Decision Letter 0]

12 Sep 2022

Dear Dr. Enciso,

Thank you very much for submitting your manuscript "Shadow enhancers mediate trade-offs between transcriptional noise and fidelity" for consideration at PLOS Computational Biology.

As with all papers reviewed by the journal, your manuscript was reviewed by members of the editorial board and by several independent reviewers. In light of the reviews (below this email), we would like to invite the resubmission of a significantly-revised version that takes into account the reviewers' comments.

Both referees point to a few major concerns with the approach and interpretation. Please update the MS considering these questions ans suggestions!

We cannot make any decision about publication until we have seen the revised manuscript and your response to the reviewers' comments. Your revised manuscript is also likely to be sent to reviewers for further evaluation.

Sincerely,

Attila Csikász-Nagy

Academic Editor

PLOS Computational Biology

Mark Alber

Section Editor

PLOS Computational Biology

Reviewer's Responses to Questions

**Comments to the Authors:**

Reviewer #1: In this work, the authors explore the noise properties and “fidelity” of sub- and superadditive enhancers, with additional commentary on the consequences of these properties for enhancer evolution. They conclude that in the additive enhancer paradigm there are no consequences for changing the number of enhancers, while both super- and subadditive enhancers offer different tradeoffs among noise, fidelity, and metabolic cost. The question is timely and the use of stochastic simulations is appropriate for this question. However, I have some concerns about the mathematical notation, choice of metrics, and dependence on parameter choice, which I find to limit the scope of the conclusions.

My first two concerns are (1) the choice of mathematical notation (which conflates different concepts) and consequently (2) the authors’ definition of fidelity, upon which many of the work’s conclusions depend. In particular, the same character R is used to denote the random variable for mRNA count, a vector of expected values for mRNA count for different production levels, and (in the summation in the equation on line 185) as the indexing variable for the sample space of expected values of mRNA count (itself a random variable?). This can lead to rather confusing statements, such as:

- “R denotes a vector composed of the mean mRNA E[R]” (lines 572-573)

- in line 183, R appears to be a random variable (as its expectation is taken in the denominator), yet in the text below (line 186) it is defined as itself an expected value

Similarly, if one writes I(X;Y), then I expect X and Y to be (vector) random variables (RVs). Yet although the authors use the notation I(R;T1), in the subsequent definition R and T1 are NOT RVs, but vectors of (apparently fixed) values.

Partly for this reason, I am skeptical of the authors’ definition and calculation of fidelity. That the moments are estimated from ODEs implies that they behave deterministically with time once the network “exists” (i.e., E[R] is not a RV; aside, it seems to be assumed but never stated that the expected values of the molecule counts are taken once the network reaches the stationary distribution). In order for E[R] to be a RV, one possible assumption is that some of the kinetic parameters can take random values that may change with each instantiation of the network (each simulation) but do not change in time once the network “exists” (making E[R] a deterministic function of a RV beta1; let’s call this bar{R}(beta1)). Indeed, this seems to be implied by the authors’ statements in lines 573-575, in which they explain that they construct R and T1 by sampling values of production rate beta1 and then, for each fixed sample value, estimating the moments of the stochastic reaction network. Yet the authors do not provide any distribution for beta1, as I would expect to be necessary for deriving a distribution for bar{R}(beta1) (or bar{T1}(beta1) or the joint distribution thereof). From this perspective, I do not understand why the authors generated histograms (binning multiple values of beta1 together (!)) to estimate the joint distribution. Moreover, defining one metric (CV) in terms of a single beta1 and another metric across multiple beta1 makes me unsure of how to actually assess the tradeoff in these two system properties, though this appears to form a large part of the authors’ concluding arguments. (This could be partially addressed by specifying that the CV is taken at some “representative” tilde{beta1}, justified in relation to the distribution of beta1.)

There are many examples in the literature where mutual information has been applied to analyze signaling in genetic networks; see, e.g., Tkacik and Walczak (2011) J Phys Condens Matter and Razo-Mejia et al. and Phillips (2020) Phys Rev E. The authors might want to look into channel capacity as a way to assess how well the mRNA levels can distinguish between different values of input, where in this case the input could be transcription factor concentration (say, T1 alone) or beta1 itself. If the authors would like to depart from this more traditional perspective, then I am open to convincing, but would need much more explanation/justification/clarification than what is present in the manuscript.

My third point of concern is that the authors have essentially only tested one parameter set (Table S.1), a point they acknowledge in the discussion. It’s good that this set was derived from experimental results, so the values should be biophysically relevant. Nevertheless, I would be more convinced at the generalizability of the conclusions if the authors had tested a few different sets of parameters to see which have the most substantial effects on noise and fidelity. It seems important to me to know, for example, how CV varies with beta1. Though a massive computational scan would be out of scope, there are also at least a few other candidate parameters whose relative values might intuitively be expected to have effects. To illustrate, consider the conclusion that changing the number of enhancers does not appear to change noise properties (lines 230-234). This would be fundamentally unsurprising if r1 were about equal to r2 (as in the authors’ simulations), since nothing else in the model exists to distinguish a TF bound to one enhancer from a TF bound to another. It is not inconceivable that a substantial difference in r1 and r2 could produce a different outcome.

Overall, the work is tackling an intriguing set of questions and the basic set of experiments is appropriate to address them. If the mathematical notation can be disambiguated and the metrics convincingly justified, then my confidence in the conclusions would be strengthened. Expanding the parameter space under consideration would then serve primarily to bolster the impact of the work by broadening the circumstances under which the conclusions are relevant.

Minor comments:

- In the main text, the authors show only one mathematical implementation each for superadditivity and subadditivity relying upon a linear relationship between kinetic rate and enhancer number (line 281, Figure 3). It should be acknowledged that with these definitions taken strictly, n above a certain value will cause kinetic rates to go negative. Also, I think the authors could consider putting Fig. S.4 in the main text, as it seems important that the noise characteristics can vary so much depending on the mathematical choice of model for implementing super-/subadditivity.

- Several paragraphs in the methods section read like tutorials. Given that many of these modeling/simulation paradigms are well established and not really the focus of the present work, I feel the level of explanation is more appropriate for the appendix than for the methods.

- The writing/figures could generally use polishing for clarity. For example:

=> lines 101-102: The way this is written could be interpreted to mean the enhancer is capable of producing different expression levels in response to different TF concentrations OR that it is capable of detecting changes in TF concentration. I assume the former, but it would be nice to have this absolutely clear.

=> “fidelity” is not defined at first usage (line 133)

=> lines 175-176: what is the “corresponding level of gene expression”?

=> line 187: could state explicitly in text that beta1 is T1 production and gamma1 is T2 production

=> Table S.1 should be referenced in lines 218-219

=> lines 242-243 and 252-255: I assume these conclusions only hold if the inputs T1 and T2 are independent? If so, please state in the text.

=> line 275: “by decreasing k_on and increasing k_off”—is it really “and” or “or”?

=> line 281: It wasn't immediately clear to me that the change was additive (as is shown in the figure). Perhaps add the mathematical expression explicitly: k_on^new = k_on - nd_1

=> Fig. 3B and 4B (left): It is very confusing to have all lines gray. I’d consider changing at least the line style to help visually distinguish the lines corresponding to each total number of binding sites.

=> Fig. 3B and 4B (right): I am not convinced a bar plot is the appropriate visualization for fidelity and noise since their quantities are not directly comparable.

=> Related to the point about mathematical notation, T_n is used to denote at various times the transcription factor, its molecule count, and the number of binding sites present for it on the enhancer. I would recommend replacing it with text in the plot axes and introducing more symbols to disambiguate elsewhere in the text as necessary.

=> Fig. 4B: What are the two datasets being plotted together (both in black)?

=> Fig. 4: It’s good to provide the values of d_1 and d_2, but they are hard to interpret when numerical values have not been provided for any of the other parameters at this point in the text.

=> line 521 repeats part of line 520

Reviewer #2: The paper by Fletcher et al provides a study of how shadow enhancers may mediate trade-offs between noise and fidelity by means of stochastic simulations and approximate results using moment-closure. This is a study with well defined questions; it is very clearly written and the results are interesting to the transcriptional noise community.

My main reservations about the paper are the accuracy of the moment-closure approximations used, the lack of investigation of the Fano factor of the system which is a common measure of transcriptional burstiness, some strong assumptions on how TF production is modelled, the lack of relevant literature cited, and the rather lengthy (though nicely written) Methods which can be much condensed by reference to review articles and books. My more detailed comments are as follows:

(1) The results of the paper (for the most part) rest upon the use of the zero cumulant moment-closure approximation which involves setting equal to 0 all the cumulants with an order greater a certain value. In the literature this has also been called “Normal moment-closure” or “cumulant neglect moment-closure”. While this has been used quite a bit, various studies have also shown its limitations and those of other moment-closure approximations. See in particular Grima, R. "A study of the accuracy of moment-closure approximations for stochastic chemical kinetics." The Journal of chemical physics 136.15 (2012): 04B616. Schnoerr et al. "Validity conditions for moment closure approximations in stochastic chemical kinetics." The Journal of chemical physics 141.8 (2014): 08B616_1. Schnoerr et al. "Comparison of different moment-closure approximations for stochastic chemical kinetics." The Journal of Chemical Physics 143.18 (2015): 11B610_1. It is important that these and similar studies are discussed since presently by reading the paper one gets the idea that moment-closure does very well but this is generally not the case and once has to be careful with the results obtained. As well, what worries me is that the authors do not show a study of the accuracy of the used moment-closure method over parameter space for this model and hence its difficult to assess its accuracy (as far as I can tell they only refer to Fig 1c to assess accuracy but this is far from an exhaustive study across param space). I suggest a more comprehensive study of the method's accuracy is shown.

(2) The authors use a moment-closure method because the system has second-order reactions and hence moments cannot be computed exactly. I note that there is an alternative method of proceeding which is based on mean-field approximation and which even gives the distributions of mRNA, not simply the moments. I am referring to the Linear Mapping Approximation (LMA) which is specifically designed for reaction systems where the 2nd order reactions involve a protein binding a gene. See the paper Cao et al. "Linear mapping approximation of gene regulatory networks with stochastic dynamics." Nature communications 9.1 (2018): 1-15. Applying this to the present system would be advantageous in that more info can be obtained without using the SSA, the amount of effort is similar to that of moment-closure but its typically more accurate since there are no implicit distribution approximations. In any case, even if they decide not to use this approach, it would be useful to discuss alternative approaches such as the LMA which can give information about the transcript number distribution which is routinely measured.

(3) The mutual information is calculated between R and T1. Why not between R and T2? The CV and the mutual information are studied. However the Fano factor (FF = variance / mean) was not. This is an important measure of transcriptional bursting since FF = 1 implies constitutive expression and deviations from 1 are a common measure of the extent of bursting; see for e.g. Sanchez et al. "Genetic determinants and cellular constraints in noisy gene expression." Science 342.6163 (2013): 1188-1193. Since the FF gives different information than the CV and since it s easy to compute from moment-closure, I strongly suggest the authors to study it.

(4) I found the Methods far too long and written for a non-expert audience. This is appropriate for a PhD thesis but given the vast amount of information present in books and published papers, I think it would be better to condense it. Reference can be made to recent review articles.

(5) The TF's are produced in bursts of fixed size n1 and n2. While I understand they may have appeared in previous publications, it is generally not the accepted way to proceed. TFs being proteins suffer from translational bursting which leads to burst size distributions that are geometric, i.e. n1 and n2 should be random numbers sampled from a geometric distribution with a certain mean burst size. There is extensive literature on this; one of the founding papers on the topic is the following Shahrezaei et al. "Analytical distributions for stochastic gene expression." Proceedings of the National Academy of Sciences 105.45 (2008): 17256-17261. A rigorous derivation is also shown here Jia. "Simplification of Markov chains with infinite state space and the mathematical theory of random gene expression bursts." Physical Review E 96.3 (2017): 032402. Implementing random bursts is also not difficult to do using Cerena. For e.g. instead of 0 -> n1 T1, we would now have a set of reactions 0 -> T1 with rate k_1, 0 -> 2 T1 with rate k_2, ..., 0 -> N T1 with rate k_N where N is some suitably large number (chosen much larger than the mean burst size) and k_i is equal to beta_1 multiplied by P(i) = probability that the burst is of size i using the geometric distribution. I anticipate that the results for the noise and fidelity will change as a result of using a geometric burst size distribution.

(6) I finally note that since TFs are proteins they are also typically stable and their degradation is not principally via an active mechanism but rather due to dilution when cell division occurs -- while I am not suggesting that they implement this for their study (since its computationally challenging) it would be useful if they can add a brief discussion of how the explicit modelling of cell division leads to larger noise than using a model where there is no explicit cell division and where rather this is taken into account implicitly via an effective rate (as done in the present model). For reference see for e.g. Beentjes, et al. "Exact solution of stochastic gene expression models with bursting, cell cycle and replication dynamics." Physical Review E 101.3 (2020): 032403.

**Have the authors made all data and (if applicable) computational code underlying the findings in their manuscript fully available?**

Reviewer #1: Yes

Reviewer #2: Yes

PLOS authors have the option to publish the peer review history of their article (what does this mean?). If published, this will include your full peer review and any attached files.

Reviewer #1: **Yes: **Mindy Liu Perkins

Reviewer #2: No
---

## [Decision Letter · Decision Letter 1]

4 Mar 2023

Dear Dr. Enciso,

Thank you very much for submitting your manuscript "Shadow enhancers mediate trade-offs between transcriptional noise and fidelity" for consideration at PLOS Computational Biology. As with all papers reviewed by the journal, your manuscript was reviewed by members of the editorial board and by several independent reviewers. The reviewers appreciated the attention to an important topic. Based on the reviews, we are likely to accept this manuscript for publication, providing that you modify the manuscript according to the review recommendations.

Reviewer 1 identified a few minor mistakes, please correct those!

Sincerely,

Attila Csikász-Nagy

Academic Editor

PLOS Computational Biology

Mark Alber

Section Editor

PLOS Computational Biology

Reviewer's Responses to Questions

**Comments to the Authors:**

Reviewer #1: The authors have addressed my major concerns about the manuscript. I have three minor comments:

1) Line 239: I assume the authors mean "the noise does NOT change" (written: "the noise does change")

2) Fig 3C: Should the x-axes be tau_1, tau_2 rather than T_1, T_2?

3) Line 448: There seem to be a couple exceptions to the "broadly consistent" trend in the superadditive case (e.g., the green curves in Fig S.6C, left and the pink curves in Fig S.6C, center/right reverse or at least do not follow the trend from the original parameter set). Given the expected audience of this publication, I think it is important to be explicit that it is possible to find parameter sets for which the observed trends do not hold, even if the authors do not expect these cases to occur in most biologically relevant situations.

Reviewer #2: The authors have replied to all my questions and done substantial changes. I am happy to recommend the manuscript for publication in PLoS Computational Biology.

**Have the authors made all data and (if applicable) computational code underlying the findings in their manuscript fully available?**

Reviewer #1: Yes

Reviewer #2: Yes

PLOS authors have the option to publish the peer review history of their article (what does this mean?). If published, this will include your full peer review and any attached files.

Reviewer #1: **Yes: **Mindy Liu Perkins

Reviewer #2: No

Figure Files:

Data Requirements:

Reproducibility:

References:

---

## [Editor Report · Decision Letter 2]

3 Apr 2023

Dear Dr. Enciso,

We are pleased to inform you that your manuscript 'Shadow enhancers mediate trade-offs between transcriptional noise and fidelity' has been provisionally accepted for publication in PLOS Computational Biology.

Best regards,

Attila Csikász-Nagy

Academic Editor

PLOS Computational Biology

Mark Alber

Section Editor

PLOS Computational Biology

---

## [Editor Report · Acceptance letter]

5 May 2023

PCOMPBIOL-D-22-01195R2 

Shadow enhancers mediate trade-offs between transcriptional noise and fidelity

Dear Dr Enciso,

I am pleased to inform you that your manuscript has been formally accepted for publication in PLOS Computational Biology. Your manuscript is now with our production department and you will be notified of the publication date in due course.

With kind regards,

Anita Estes
